# Insights into Hepatocellular Carcinoma in Patients with Thalassemia: From Pathophysiology to Novel Therapies

**DOI:** 10.3390/ijms241612654

**Published:** 2023-08-10

**Authors:** Pei-Chin Lin, Wan-Yi Hsu, Po-Yi Lee, Shih-Hsien Hsu, Shyh-Shin Chiou

**Affiliations:** 1Division of Pediatric Hematology and Oncology, Department of Pediatrics, Kaohsiung Medical University Hospital, Kaohsiung 807378, Taiwan; cooleylin@gmail.com (P.-C.L.); silverglassfish@gmail.com (W.-Y.H.); leeboylle@gmail.com (P.-Y.L.); 2School of Post-Baccalaureate Medicine, College of Medicine, Kaohsiung Medical University, Kaohsiung 807378, Taiwan; 3Graduate Institute of Medicine, College of Medicine, Kaohsiung Medical University, Kaohsiung 807378, Taiwan; 4Center of Applied Genomics, Kaohsiung Medical University, Kaohsiung 807378, Taiwan; 5Graduate Institute of Clinical Medicine, College of Medicine, Kaohsiung Medical University, Kaohsiung 807378, Taiwan; 6Division of Laboratory Medicine, Kaohsiung Medical University Hospital, Kaohsiung 807378, Taiwan

**Keywords:** hepatocellular carcinoma (HCC), thalassemia, iron, transfusion-dependent thalassemia (TDT), non-transfusion-dependent thalassemia (NTDT), reactive oxygen species (ROS)

## Abstract

Thalassemia is a heterogeneous congenital hemoglobinopathy common in the Mediterranean region, Middle East, Indian subcontinent, and Southeast Asia with increasing incidence in Northern Europe and North America due to immigration. Iron overloading is one of the major long-term complications in patients with thalassemia and can lead to organ damage and carcinogenesis. Hepatocellular carcinoma (HCC) is one of the most common malignancies in both transfusion-dependent thalassemia (TDT) and non-transfusion-dependent thalassemia (NTDT). The incidence of HCC in patients with thalassemia has increased over time, as better chelation therapy confers a sufficiently long lifespan for the development of HCC. The mechanisms of iron-overloading-associated HCC development include the increased reactive oxygen species (ROS), inflammation cytokines, dysregulated hepcidin, and ferroportin metabolism. The treatment of HCC in patients with thalassemia was basically similar to those in general population. However, due to the younger age of HCC onset in thalassemia, regular surveillance for HCC development is mandatory in TDT and NTDT. Other supplemental therapies and experiences of novel treatments for HCC in the thalassemia population were also reviewed in this article.

## 1. Introduction

Thalassemia is a heterogeneous congenital hemoglobinopathy common in the Mediterranean region, Middle East, Indian subcontinent, and Southeast Asia [1,2,3,4]. Immigration has led to the gradual increase in the incidence of thalassemia in Northern Europe and North America [5,6]. Thalassemia can be classified based on defects in the globin chain, the α-globin chain or β-globin chain as α-thalassemia or β-thalassemia, respectively. Thalassemia is the most common monogenic disease worldwide, with approximately 1.5% of the population carrying the β-thalassemia allele and 5% carrying the α-thalassemia allele [5,7]. Clinical manifestations include anemia, jaundice, and iron overload, and the disease severity ranges from near-normal without complications to requiring lifelong transfusion support [3,6]. Patients with thalassemia requiring transfusion support are classified into transfusion-dependent thalassemia (TDT) and non-transfusion-dependent thalassemia (NTDT) based on the frequency and regularity required. The lifespan will be less than 10–15 years without transfusions, and with transfusion therapy, it can be extended to approximately 50 years depending on the management of the iron overloading resulting from repeated red cell transfusions, which damage organ functions, especially the heart, liver, lung, pancreas, and pituitary glands [3,8,9]. Besides transfusions, increased dietary iron absorption is one of the major causes of liver iron overloading, typically in patients with NTDT [6,8]. Liver iron deposits lead to liver fibrosis, cirrhosis, and HCC [8]. The relationship between iron overload and HCC development is clear in that the incidence of HCC is much higher in patients with hereditary iron overload than in the general population [10]. A prospective study found that the incidence of HCC in patients with a beta-thalassemia major was approximately the same as the risk for HCC in the general population but with a significantly younger age at HCC diagnosis [11]. The incidence of HCC in patients with thalassemia has increased over time as better chelation therapy confers a sufficiently long lifespan for HCC development [12]. According to an analysis based on the Taiwan Health Insurance Longitudinal Database (1998–2010), patients with thalassemia had a significantly higher risk for abdominal cancer (aHR = 1.96, 95% CI 1.22–3.15) compared with the comparison group. Patients with thalassemia who received blood transfusions were 9.12 times more likely to develop abdominal cancer than those who did not receive blood transfusions [13]. Iron accumulation can also lead to various endocrinopathies, such as hypogonadism, impaired pancreatic excrete function, and hypothyroidism [14]. This review discusses the pathogenesis, molecular mechanisms, management, and perspectives of iron overloading associated with HCC in patients with thalassemia.

## 2. Risk Factors for HCC

### 2.1. Risk Factors for HCC in the General Population Focusing on Hepatitis Virus and Fatty Liver Disease

HCC is the sixth common cancer worldwide and the third leading cause of cancer death. The incidence in the general population varies widely in different world regions, with the highest in Mongolia (85.6 per 100,000) and the lowest in Sri Lanka (1.2 per 100,000) [15]. Before the advent of a vaccine against hepatitis B virus (HBV), the hepatitis B and C viruses (HCV) were the main risk factors for HCC over the past 40 years. The risk of HCC development in HBV carriers ranges from 9.6% in Hepatitis B surface antigen (HBsAg) positive patients to 60% in double positive (HBsAg + Hepatitis B e antigen (HBeAg)) patients [16]. In HCV-related cirrhosis patients, the risk of progression to HCC is estimated to be 2–6% each year [17].

Today, with a modern high-fat/sugar diet, sedentary lifestyle, and increasing alcohol consumption, alcoholic liver disease (ALD) and non-alcoholic fatty liver disease (NAFLD) are gradually being recognized as risk factors for HCC. Alcohol-related cirrhosis accounted for about 15–30% of HCC cases in one report [18]. The pathogenesis from ALD to HCC is complex; liver injury from the insult of reactive oxygen species (ROS) and acetaldehyde are contributing factors [16]. In chronic viral hepatitis patients, alcohol intake is associated with higher all-cause mortality and increases HCC development risk compared to the no alcohol intake group [19,20]. The global prevalence of NAFLD is estimated about 25% [21], and it is recognized now as the most common cause of cirrhosis and a growing risk factor for HCC [22]. Oxidative stress, chronic inflammation, and immune cell infiltration of fatty liver are all factors leading to HCC formation [23].

### 2.2. Risk Factors for HCC in the Thalassemia Group Focusing on Hepatitis Virus and Fatty Liver Disease

Because of life-long and frequent blood transfusions, thalassemic patients have a higher incidence of acquiring transfusion-transmitted viral infection, especially HBV and HCV. HBsAg positive was estimated in 0.3~5.7% of thalassemic patients [24]. Anti-HCV Ab was estimated 4.4~85.4% [25]. The actual incidence may vary widely because of differences in regional HBV/HCV prevalence.

As mentioned in the introduction, thalassemia major patients had impaired pancreatic function because of iron overload. The prevalence of diabetes mellitus (DM), mostly type 1, and impaired fasting sugar in thalassemic patients was about 6.54% and 17.21%, respectively [26]. NAFLD was deemed to be the result of insulin resistance and metabolic syndrome in the past, but now a high prevalence in type 1 DM patients is noted. Altered dynamic insulin pulsatile, altered insulin clearance, hyperglucagonemia, and hepatic glucagon resistance are parts of NAFLD pathogenesis in type 1 DM. These are slightly different from type 2 DM and metabolic syndrome patients [27]. Although there are no formal statistics of NAFLD prevalence in thalassemic patients, we inferred that owing to type 1 DM and iron overload (discussed next), the prevalence of NAFLD would be high in thalassemic patients.

### 2.3. Risk Factors for HCC in the Thalassemia Group Focusing on Iron Overload

In addition to the abovementioned risk factors, a special concern regarding HCC in the thalassemia group is iron overload resulting from lifelong blood transfusion and ineffective hematopoiesis. The NTDT group also suffers from iron overload to a lesser degree because of excessive hemolysis and chronic anemia. Iron overload is mainly due to blood transfusions and an excess of destroyed red blood cells (RBCs). To make things worse, patients with thalassemia generally have low hepcidin levels because of chronic anemia; so, their enterocytes still absorb iron greedily, even under an iron overload status [28].

Iron, an essential element for most species to live, is the most abundant metal on our planet [29]. Iron can easily gain or lose electrons by switching between the iron form (Fe^2+^) and the iron form (Fe^3+^). This characteristic enables iron to participate in the respiratory chain reaction, electron transfer, and signal transduction in cells and functions as an oxygen carrier in hemoglobin and myoglobin. The human body absorbs iron from food through enterocytes, which mostly go to the bone marrow for hemoglobin formation. RBCs are the largest iron reservoir in our body. After meeting the need for basic cellular function, surplus iron is mainly stored in the liver. Iron is such a precious metal for humans that we have not evolved the ability to excrete iron out of our body actively. Instead, we decrease our iron store reluctantly (passively) from blood loss. Menstruation is the only nontraumatic and natural way of losing blood [30,31]. To summarize, it is easy to accumulate iron in the body and almost impossible to excrete it, except during menstruation in women.

From the cellular level, various iron importers exist on the cell membrane, such as transferrin receptor protein 1 (TfR1), transferrin receptor 2 (TfR2), L-type calcium channel (LTCC), divalent metal transporter 1 (DMT1), zinc transporter 8 (ZIP8), and transient Receptor Potential Cation Channel Subfamily C Member 6 (TRPC6); however, ferroportin is the only exporter [30,31]. When thalassemia patients are forced to receive iron (ex. Blood transfusion), there is no way for them to excrete excessive iron. In the long-run, more and more iron participates in the body, mainly in the liver, and causes organ damage and affects cell function profoundly.

Iron overload is a risk factor for HCC. Hereditary hemochromatosis patients were more likely to have liver cancer compared to the general population [32,33]. It is estimated that iron-overloaded patients have a 10.6 times greater risk for HCC and even develop HCC before cirrhosis development [34]. Furthermore, hepatitis virus and iron can interact with each other and aggravate the liver cirrhosis process and HCC formation [35].

## 3. Pathogenesis of Viral Infection and NAFLD with HCC

The pathogenesis of HCC in patients with thalassemia is more complicated than in the general population. Figure 1 delineates the possible risk factors and associations between conditions specific to thalassemia and the development of HCC. The interactions of HBV/HCV, metabolic syndrome, and HCC are mentioned briefly in this section. The pathogenesis of iron-deposit-associated HCC is discussed separately in the next section.

### 3.1. HBV/HCV-Associated HCC

HBV and HCV contribute to HCC through different mechanisms. HBV is a DNA virus. It can integrate into the host genome and promote HCC formation by causing chromosome instability, promoting oncogene expression and activating cancer signaling pathways [36,37]. In an HCC patient with an HBV infection history, around 85% to 90% of the tumor tissue contains HBV DNA [38]. HBV infection also induces epigenetic dysregulation. It affects gene methylation and histone modification, activating oncogenes and silencing tumor suppressor genes. HBV is also capable of regulating microRNAs’ expression to advocate tumor metastasis, drug resistance, and proliferation. HBV infection also modulates our innate and adaptive immune system and transitions the liver microenvironment to a immunosuppressive status [36].

HCV is an RNA virus; so, it cannot integrate into the host genome consistently. Therefore, HCV infection does not have a strong oncogenic effect on HCC formation as HBV does, and it has also been less studied compared to HBV. HCV infection promotes HCC formation mainly by inducing epigenetic dysregulation, altering the cellular signaling pathway through viral RNA-encoded proteins [39].

### 3.2. NAFLD/NASH-Associated HCC

The process from NAFLD to HCC is deemed as a continuous one of steatosis [so-called non-alcoholic steatohepatitis (NASH)] with chronic inflammation to cirrhosis to carcinogenesis. About 2–13% of NAFLD-related cirrhosis patients have HCC development. However, around 8% of NAFLD-related HCC cases skip the cirrhosis stage and evolve to cancer directly [40,41]. Today, the process from NAFLD to NASH to cirrhosis is thought to be due to multiple factors, including chronic inflammation from free fatty acid and insulin resistance status, endotoxin and organokines from liver, altered immune cells, and the influence of gut-derived endotoxins [42,43]. These factors cause tumorigenesis as well.

Cirrhosis provides a microenvironment with a cancer-filled effect, where neoangiogenesis, fibrosis, and depressed immune surveillance offer senescent hepatocytes or primitive hepatocytes with mutated genes a chance to proliferate [44,45]. However, some studies have shown that cirrhosis restricts HCC development and can be an additional factor of immunosurveillance [46]. Thus, whether cirrhosis is a true risk factor for HCC is still under debate.

## 4. Pathogenesis of Iron Overload and HCC

### 4.1. Iron and Inflammation

When intrahepatic iron overflows, hepatocytes are also stimulated to differentiate into a proinflammatory state through excessive intracellular ferritin by enhancing the NF-κB expression pathway [47]. Prolonged activation of hepatocytes means they can differentiate into myofibroblasts and lead to neovascularization and extracellular matrix formation. Excessive iron molecules are not only deposited in but also injure hepatocytes. They also affect immune system function. Liver, as the largest reticuloendothelial system in the body, contains the largest population of macrophages (Kupffer cells). Macrophages, as scavengers in the body, phagocytize not only external harmful agents, such as bacteria, but also old dysfunctional cells, such as senescent RBCs and destroyed hepatocytes. Thus, they are prone to contain excessive amounts of iron, especially in patients with thalassemia. Excessive amounts of iron induce the macrophage to secrete proinflammatory cytokines, such as tumor necrosis factor-alpha (TNF-α) and interleukin-12 (IL-12) [48]. Together, with the proinflammatory status of hepatocytes, complicated chronic inflammatory interaction ensues. Hepatic stellate cells, endothelial cells, and lymphocytes inside the liver all participate, and cirrhosis begins [49]. Cirrhosis provides an oncogenic environment for HCC formation [50,51,52]. A study of human liver tissue revealed that hepatocytes were polarized to myofibroblasts when the liver iron concentration (LIC) was >60 μmol/g. Cirrhosis ensues when the LIC is >250 μmol/g [53]. Fibrosis develops when the LIC is >400 μmol/g [54].

### 4.2. Iron and Reactive Oxygen Species (ROS)

When intracellular iron is excessive, these iron molecules participate in the Fenton reaction (Figure 2) and cause the accumulation of ROS [55]. Healthy cells can detoxify ROS through nonenzymatic and enzymatic antioxidants, such as glutathione, flavonoids, superoxide reductase, catalase, and glutathione peroxidase. If ROS formation exceeds the detoxifying ability of cells, excess ROS can cause genomic instability and reprogram cell metabolism toward carcinogenesis [56]. ROS induce hypoxia-inducible factor-1α expression and related angiogenic gene expression and thus promote angiogenesis [57]. ROS can also promote ERK/NF-kB pathway activation and facilitate tumor cell growth, migration, and deposition of extracellular matrix [58]. To summarize, ROS enhance tumor cell proliferation and survival.

Other than iron, HBV/HCV infection and NASH/NAFLD status provide many oxidative stresses to hepatocytes as well. In NAFLD, elevated mitochondrial fatty acid oxidation and impaired respiratory chain activity make ROS accumulate [59]. It was found that the HBV structural protein-HBx damaged the mitochondrial membrane potential by interacting with the outer membrane and affecting the respiratory complex; ROS production then increased [60]. With chronic HCV infection status, patients usually have a higher liver iron content and thus high ROS production [61]. Thus, in thalassemia patients, co-existence of hepatic viral infection and/or NAFLD with iron overload makes them vulnerable to hepatocarcinogenesis.

### 4.3. Hepcidin and Ferriportin

Hepcidin is a polypeptide hormone, isolated from plasma and urine samples with antimicrobial activities by two research groups in 2000 [62,63]. They showed that hepcidin was mainly produced by the liver, followed by the heart, brain, and lung [62]. Almost at the same time, Pigeon et al. tested the hepatic gene expressions in iron-overload mouse models and showed that hepcidin (HEPC), the gene encoding hepcidin, was overexpressed in iron overload hepatic cells. In the report, they also mapped the gene to the human chromosome 19 close to the upstream stimulatory factor-2 (USF2) gene [64]. Lipopolysaccharide, a main component of pathogenic molecules, acts on macrophages, including hepatic Kupffer cells, to induce interleukin-6 (IL-6) production, which in turn induces the overexpression of hepcidin mRNA in hepatocytes [65,66]. In summary, iron overloading, infection, and inflammation can induce hepatocyte hepcidin secretion, which leads to hypoferremia and the anemia of chronic inflammation. Anemia and hypoxia suppress hepcidin expression and result in tissue iron overload. Hepcidin regulates the absorption of dietary iron from the intestine, the release of recycled hemoglobin iron by macrophages, and the movement of stored iron from hepatocytes by blocking the iron transport in the intestinal epithelium, the placenta, and the macrophage [65]. Ferriportin, the only ion exporter in human cells, plays a critical role in hepcidin-regulated iron metabolism, where ferriportin–hepcidin binding leads to the internalization and degradation of ferriportin, and due to the decreased amounts of ferripotin, iron is trapped in hepatocytes, macrophages, and absorptive enterocytes. In consequence, cytoplasmic iron accumulates and reduces the cell uptake of iron. After plasma iron becomes depleted, mainly through hemoglobin synthesis by red cell precursors in the bone marrow, the hepcidin expression is suppressed and restores the iron hemostasis system [67].

As a key regulator of iron metabolism, hepcidin levels are influenced in the pathogenesis of anemia. Inefficient erythropoietic anemias, such as thalassemia, whose clinical manifestation is iron overload anemia, iron deficiency anemia, and congenital erythropoiesis, often have low or abnormal hepcidin relative to the anemia [28]. Kijima et al. examined hepcidin expression in cancerous and noncancerous liver tissues from 40 HCC patients and found that hepcidin mRNA expression was significantly suppressed in cancerous tissues compared to noncancerous liver tissues. It is noncancerous in HCC patients, regardless of the disease state such as tumor differentiation or the time to relapse. There was no significant difference in the mRNA expression of ferripotin and transferrin receptor 2 (TfR2) between cancerous and noncancer tissues [68]. On the other hand, Tan et al. found that hepcidin, TfR2, transferrin (Tf), ceruloplasmin (Cp), and iron regulatory protein 1 (IRP1), isolated in 24 HCC patients with chronic HBV infection, were downregulated compared to the adjacent tumor-free liver tissue and normal liver controls [69]. Similarly, Tseng et al. found that hepcidin, ceruloplasmin, transferrin, and transferrin-phosphorus receptor 2 were downregulated, transferrin receptor 1 was upregulated, and the hepcidin levels consistently correlated with hepatic iron stores in 50 HCC patients [70]. Kessler et al. showed that in cirrhotic tissues, hepcidin expression was lower compared to healthy liver samples in an HBV-related cohort as well as in HCV-infected patients, hepcidin expression was even lower in the HCC samples, and in their mouse models, they showed that hepcidin expression was decreased in early hepatocarcinogenesis as well as in a later stage of murine tumorigenesis [71]. Hepcidin also ameliorates liver fibrosis by inhibiting hepatic stellate cells (HSCs)’ activation, which facilitates liver fibrosis via the degradation of ferroportin in HSCs, increasing Akt phosphorylation and finally prohibiting TGFb1-inducible Smad3 phosphorylation [72].

The underlying mechanisms of suppressed hepcidin in HCC included the suppression of *HAMP*, *TfR2*, *HJV*, *ALK2*, and/or circular RNA circ_0004913, upregulations of matriptase-2 and/or GDF15, and the inactivation of *RUNX3* and/or mutations in *TP53* [73]. Hepcidin-induced ferriportin dysregulation has been reported to be associated with an increased risk of cancer, including HCC. FPN overexpression has been reported to be associated with a significant reduction in clonogenic ability, tumor regression, and liver metastasis in breast cancer cells [74,75]. The effects of hepcidin downregulation involve increased cancer proliferation via activation of the CDK1/STAT3 pathway and JAK/STAT pathway in a dependent manner [73,76].

## 5. Management

### 5.1. Management of Iron Overload

Once iron overloaded, patients with thalassemia should receive iron-chelation therapy. There are three iron-chelating drugs commercially available, deferoxamine (DFO) in subcutaneous or intravenous form, deferiprone (DFP) in oral form, and deferasirox (DFX) in oral form. All three drugs are useful for TDT patients, while DFX remains the only approved drug for NTDT patients. Selection of the iron chelator should be on the basis of the patient age, iron overload profile, presence or absence of comorbidities, patient preference and adherence, and side effects. For TDT patients with serum ferritin ≥ 1000 ng/mL or after transfusion of 10 units of packed red blood cells, iron-chelation therapy is recommended. For NTDT patients, iron-chelation therapy starts when serum ferritin ≥ 800 ng/mL or LIC ≥ 5 mg/g [9,77].

Iron overload and increased non-transferrin-bound iron lead to ROS production and subsequent oxidative damage. It is reasonable to assume that antioxidant therapy will be beneficial to iron overloaded thalassemia patients. Encouraging results from several randomized controlled trials have shown that natural antioxidants such as green tea and curcumin may be helpful; however, studies with a longer duration are needed to assess the long-term effects [78,79,80,81]. In 2021, according to the Thalassemia International Federation guidelines for the management of TDT, a diet rich in foods high in vitamin E, such as plant oils, eggs, nuts, and cereals, is recommended. To support the prolonged use of vitamin E supplements, further research is still needed [82].

### 5.2. Management of HCC

Because of limited data, the treatment of HCC in patients with thalassemia is primarily based on experience in non-thalassemic patients. Current treatment recommendations for HCC in the general population follow the Barcelona Clinic Liver Cancer (BCLC) staging system, in which patients with HCC are classified into five stages (stage 0: very early HCC, stage A: early HCC, stage B: intermediate HCC, stage C: advanced HCC, and stage D: terminal HCC), according to several prognostic variables, including tumor status, liver function (defined by Child-Pugh class), and performance status. In principle, patients with stage 0 and stage A should undergo surgical resection if they are candidates or local ablation. Liver transplantation is also a potentially curative treatment option for unresectable disease if eligible. For patients with advanced disease, systemic therapies will be applied first [83,84,85].

Patients with intermediate-stage HCC are highly heterogeneous in terms of tumor burden and liver function, and only a subset of this population could benefit from transarterial chemoembolization (TACE). To prolong overall survival, it is crucial to select the right patients for TACE or systemic therapy. In the 2019 Asia–Pacific Primary Liver Cancer Expert (APPLE) consensus statements, systemic targeted therapy was recommended as the initial treatment for intermediate-stage HCC patients with high tumor burden, who were unsuitable for TACE. The up-to-7 criteria is the most commonly used criteria to define a high tumor burden. A patient is beyond up-to-7 criteria if the sum of the tumor number and the diameter of the largest tumor (in centimeters) exceeds 7 [86]. In 2021, Huang et al. from Taiwan further classified intermediate-stage HCC patients by the newly proposed 7–11 criteria, showing a better performance in predicting radiologic response and survival after TACE [87].

The treatment modalities that have been reported and shown to be safe for HCC in thalassemia patients include surgical resection, radiofrequency ablation, microwave ablation, percutaneous ethanol injection, chemoembolization, radioembolization, sorafenib, and liver transplantation [11,24,88,89,90,91,92,93].

For a long time, liver transplantation has been refused to thalassemia patients mainly due to cardiac comorbidities. Based on the success stories reported in a series of cases, thalassemia per se should no longer be considered a contraindication to liver transplantation [24,93]. Liver transplantation is a potential treatment option for thalassemia patients once there are no significant comorbidities, such as severe pulmonary hypertension and overt heart disease [12,77,94,95].

In 2007, sorafenib became the first approved systemic therapy for HCC. Currently there are many approved therapies for advanced HCC, such as lenvatinib, regorafenib, cabozantinib, ramucirumab, nivolumab, pembrolizumab, the combination of atezolizumab and bevacizumab, the combination of nivolumab and ipilimumab, and the combination of tremelimumab and durvalumab [83,84,85,96,97,98]. Published data are lacking on use of these agents in patients with thalassemia and HCC. However, treatment of HCC in thalassemia patients should follow the same recommendations as in the non-thalassemic population. Novel immunotherapies, combination therapies, and chimeric antigen receptor (CAR)-T cell therapies are being tested for HCC, and thalassemia should not be regarded as a contraindication.

Intervention or modalities intending to block or reverse liver fibrosis are other possible treatment policies in patients with liver fibrosis before cancer transformation occurs. Hepatic stellate cells (HSCs)’ activation is a critical process in the development of liver fibrosis. HSCs reside in the space of Disse and are usually quiescent. Chronic inflammation of hepatocytes will activate HSCs, which then transform to myofibroblast-like cells. In the extracellular matrix (ECM) in fibrotic liver tissue, activated HSCs are the major components leading to excessive ECM deposition and scarring [99]. Chronic inflammation and liver fibrosis also create a microenvironment that can promote hepatocyte carcinogenesis by inducing genetic mutations and genomic instability in hepatocytes, influencing cell signaling pathways, angiogenesis, and immune responses. Activation of the TGF-β and Wnt/β-catenin signaling pathways and the aggregation of immune cells, such as tumor-associated macrophages can promote tumor growth and angiogenesis [100]. Significantly increased mitogenic cytokine IL6 and TNF levels were noted in advanced fibrosis, leading to a propensity towards cancer by regulating immune cells and the growth of tumor cells [101]. Hepatic fibrosis was previously recognized irreversible, but sequential liver biopsies in hepatitis B, hepatitis C, and NASH showed that removing the etiologies has a chance to reverse hepatic fibrosis [99]. Increased apoptosis or inactivation of HSCs is an important mechanism for the resolution of liver fibrosis [102]. Antifibrotic therapies that mediate its antifibrotic effects by hepatocyte protection, inhibition of HSC activation and fibrotic scar evolution or immune modulation may have therapeutic effects in liver fibrosis [102]. Clinical trials using Pan-caspase inhibitor, ASK1 inhibitor, Dual NOX1/4 inhibitor, FXR agonist, CBP/β-catenin small molecule inhibitor, Hsp47 siRNA delivering lipid nanoparticle, LOXL2 specific monoclonal antibody, CCR2/CCR5 inhibitor, and Inhibitor of galectin-3 were conducted for management of liver fibrosis with various outcomes [102]. Notably, Se, which is an important trace element metabolized in liver, can reduce oxidative stress, inhibit tumor growth, and prevent liver damage. The literature has shown that an increase in serum Se leads to an increase in the expression of selenoprotein P, and the inhibition of proinflammatory factors and its content in liver cancer cells is significantly reduced compared to healthy ones [103]. Selenium nanoparticles have shown good results in a variety of liver pathologies (inflammation, infectious, and oncological diseases) suggesting their hepatoprotective and anticancer effects both in vitro on liver cell culture and in vivo on animal models [103].

## 6. Prognosis

Worldwide, liver cancer is the sixth most common cancer and the third leading cause of cancer-related death. HCC accounts for 75–85% of liver cancer cases [15]. Thanks to advances in early diagnosis and treatment, survival rates for HCC have increased over the past three decades, with Surveillance Epidemiological Final Results reporting an overall increase in the 1-year survival rates from 18.2% to 51.2% [104]. The recently reported median survival of HCC in the general population was approximately 6.1–10.3 months. Survival differences were noted across etiologies of HCC due to differential screening and treatment practices, with more favorable survival in HBV-related HCC [95]. More studies have focused on HCC in patients with thalassemia; however, survival data for this population are still limited. In 2004, Borgna-Pignatti et al. from Italy reported that based on data collected from 52 centers, the median survival after HCC diagnosis was 3.5 months in patients with TDT and HCC [88]. Ten years later, in their update report, the median survival rose to 11.5 months [24]. In a previous study, the prognostic factor of survival in patients with TDT and HCC was investigated, and the only significant factor was Barcelona Clinic Liver Cancer staging [89].

## 7. Surveillance

In patients with thalassemia, iron overload and liver fibrosis must be monitored, and early-onset HCC detected. For the measurement of LIC, magnetic resonance imaging (MRI) T2* or R2* is a validated tool and has replaced invasive liver biopsy in current clinical practice. The current surveillance program suggests that the frequency of MRI should be based on the baseline LIC. If the baseline LIC is <3 mg/g, MRI is encouraged every 2 years; if 3–15 mg/g, MRI is recommended annually; and if >15 mg/g or the LIC or serum ferritin is rapidly increasing, MRI is advised every 6 months [105]. Transient elastography (e.g., Fibroscan^®^) is a reliable and accurate noninvasive test to assess liver fibrosis. According to current practice, transient elastography should be performed every 6–12 months if available [105,106].

Surveillance programs for HCC aim to detect cancer at early stages in patients at a high risk, for which potentially curative treatment options are indicated. The proposed mean doubling time of an HCC lesion is 117 days [107]. Current guidelines from the American Association for the Study of Liver Disease recommend biennial followups with abdominal ultrasonography, with or without alpha-fetoprotein (AFP). Considering cost-effectiveness and safety, multiphasic computed tomography (CT) and multiphasic MRI are reserved for diagnostic evaluation despite their high diagnostic efficiency [85]. In a previous meta-analysis, ultrasonography combined with AFP provided 63% sensitivity, higher than the 45% sensitivity with ultrasonography alone [108]. To improve the sensitivity for early detection of HCC, further studies may clarify the role of advanced imaging as a screening modality in selected cases.

Besides AFP, several new biomarkers and models such as the GALAD (gender, age, lectin-reactive AFP, AFP, des-gamma-carboxy prothrombin) score are being evaluated [109,110,111]. Novel biomarkers assays, such as cell-free DNA and microRNA, are also promising [110,111,112,113,114]. However, the optimistic results come from early-phase biomarker studies. New biomarkers need to be evaluated and validated in further studies before they can be incorporated into routine clinical practice.

HCC surveillance in thalassemia using abdominal ultrasonography, with or without AFP, has been proposed in the literature. The frequency of the majority of surveillance programs is every 6 months. However, differences exist regarding the target population. Some authors have suggested screening for HCC in all patients with thalassemia, while other authors have suggested screening in those with risk factors for HCC, including serum ferritin ≥ 1000 ng/mL, LIC ≥ 5 mg/g in NTDT, LIC ≥ 7 mg/g in TDT, concurrent HBV and/or HCV infection, and advanced cirrhosis [12,77,95]. In one review, the authors recommended screening in thalassemia patients aged > 30 or younger patients if a risk factor was present, given that the youngest patient reported to have thalassemia with HCC was 33 years old [106].

## 8. Conclusions

HCC is one of the most common cancers worldwide and the third leading cause of cancer death. In patients with thalassemia, repeated transfusion and increased gastrointestinal absorption of iron lead to liver iron deposit, which in turn result in ROS accumulation and proinflammatory pathway activation. Under the chronic inflammation status, cirrhosis gradually develops and becomes an oncogenic circumstance of hepatocytes. Therefore, HCC is one of the major complications in patients with thalassemia, and management for iron chelation, regular surveillance, and other supplementary treatments, like antioxidant therapy, were recommended. Treatment of HCC in patients with thalassemia is based on that in non-thalassemic patients, except that more consideration should be given to the cardiac and pulmonary conditions that are also influenced by iron overload. In the future, new treatment modalities, such as Luspatercept and gene therapy for thalassemia, can reduce the transfusion requirement and the iron deposit, and may be beneficial for preventing HCC development in patients with thalassemia.

## Figures and Tables

**Figure 1 ijms-24-12654-f001:**
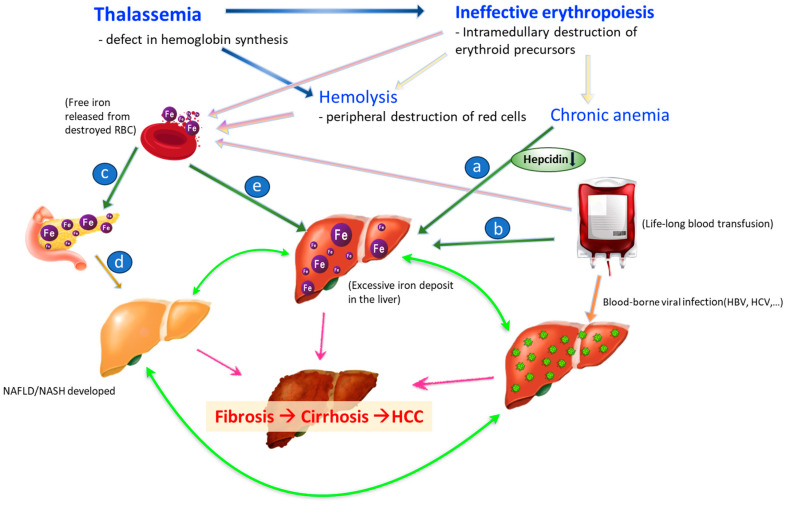
Mechanisms of HCC development in thalassemia. a (in blue circle): Thalassemia is characterized by ineffective erythropoiesis and has low hepcidin levels and iron overload. b (in blue circle): Excessive iron from senescent RBC. c (in blue circle): Excessive iron deposit in endocrine organ results in pancreatic exocrine insufficiency which causing type 1 DM. d (in blue circle): high fat/fructose/glucose diet, and sedentary lifestyle. e (in blue circle): Excessive iron also deposits into liver. Blue arrow: Thalassemia causes premature destruction of RBC precursor and hemolysis owing to deformed RBC. Light pink arrow: Hemolysis, ineffective hematopoiesis, and chronic transfusions lead to premature death of RBC. Free iron is released accordingly. Bright pink arrow: NASH, iron overload and chronic hepatitis viral infection all accelerate the process of fibrosis to cirrhosis and worst but not always consequential, to HCC. Green arrow: NASH, iron overload and chronic hepatitis viral infection can aggravate each other and make the progress of liver fibrosis early-onset, more inevitable, and severe.

**Figure 2 ijms-24-12654-f002:**
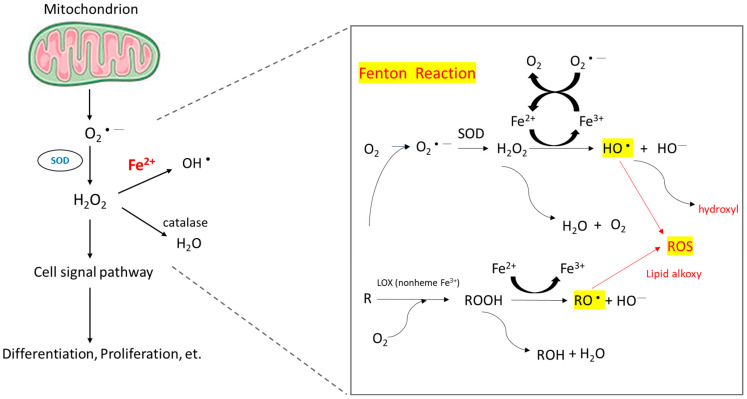
Mechanism of ROS accumulation caused by the Fenton reaction. The reaction of peroxides with Fe^2+^ to yield soluble hydroxyl (HO•) or lipid alkoxy (RO•) radicals is referred to as the Fenton reaction.

## Data Availability

Not applicable.

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
