# Peer review of "Insights into Hepatocellular Carcinoma in Patients with Thalassemia: From Pathophysiology to Novel Therapies"

_ijms, 2023, doi:10.3390/ijms241612654_

Round 1
Reviewer 1 Report
1) 2.2 Risk factors for HCC in the thalassemia group: focus on hepatitis virus, fatty liver disease. This section needs to be expanded. Data on molecular mechanisms are needed. 2) 2.2 Risk factors for HCC in the thalassemia group: focus on hepatitis virus, fatty liver disease. Schematic required for this section 3) The stages of fibrosis should be discussed. The transition of fibrosis to carcinogenesis. Including molecular mechanisms. IJMS | Free Full Text | The Role of Selenium Nanoparticles in the Treatment of Liver Pathologies of Various Natures (mdpi.com) 4) 3.2 Iron and reactive oxygen species (ROS). The role of ROS in the genesis of hepatocellular carcinoma has not been fully elucidated. This section needs additional information and more detailed analysis. 5) I suggest that the authors introduce a section discussing the data presented.
The quality of the English is good, but the text needs additional proofreading for typos.
Author Response
Reviewer 1
- 2.2 Risk factors for HCC in the thalassemia group: focus on hepatitis virus, fatty liver disease. This section needs to be expanded. Data on molecular mechanisms are needed.
Response: We thank the reviewer for this pertinent suggestion. Accordingly, we have revised the manuscript extensively to incorporate the reviewers’ suggestions. Two paragraphs on the pathogenesis and molecular mechanisms of viral infection and NAFLD induced HCC are provided in the Section 2.2-3 and Section 3. (p3-5)
- 2.2 Risk factors for HCC in the thalassemia group: focus on hepatitis virus, fatty liver disease. Schematic required for this section.
Response: We thank the reviewer’s suggestion. Accordingly, we have revised the Section 2.2-3 to incorporate the reviewers’ suggestions. Further, a figure (Figure 2) is added to illustrate the potential mechanisms of HCC in thalassemia. (p4)
- The stages of fibrosis should be discussed. The transition of fibrosis to carcinogenesis. Including molecular mechanisms.
Response: We thank the reviewer’s suggestion. Accordingly, we have added paragraphs on this point and cited the suggested article.
4) 3.2 Iron and reactive oxygen species (ROS). The role of ROS in the genesis of hepatocellular carcinoma has not been fully elucidated. This section needs additional information and more detailed analysis.
Response: We thank the reviewer’s suggestion. Accordingly, we have added more relevant information in the Section 4.2. (p6)
5) I suggest that the authors introduce a section discussing the data presented.
Response: We thank the editor’s suggestion. Accordingly, we have added relevant discuss content in ROS effects on the risk and therapy of HCC. (p7 and 9).

Reviewer 2 Report
Dear Authors
Your article is very interesting and well written. In 4.2 section there is in the 10 th line "For patients with stage B, embolization is the preferred treatment". Stage B is very complex and it has been divided into more substages , and chemoembolization is preferred treatment only in some of them, for example in those with " up to 7 criteria ". It sentence demans revision demands and more precise formulation.
Author Response
Review 2,
Your article is very interesting and well written. In 4.2 section there is in the 10 th line "For patients with stage B, embolization is the preferred treatment". Stage B is very complex and it has been divided into more substages, and chemoembolization is preferred treatment only in some of them, for example in those with " up to 7 criteria ". It sentence demans revision demands and more precise formulation.
Response: We thank the editor’s suggestion. Accordingly, we have revised the sentences and added a paragraph to clarify the treatment modalities for diseases between stage 0/A and advanced stage. (p8)

Round 2
Reviewer 1 Report
the article can be accepted for publication in its current form
Reviewer 2 Report
Thank you for your response and corrections in the manuscript.